# Approaches and Challenges in Characterizing the Molecular Content of Extracellular Vesicles for Biomarker Discovery

**DOI:** 10.3390/biom14121599

**Published:** 2024-12-14

**Authors:** Suman Kumari, Christopher Lausted, Kelsey Scherler, Alphonsus H. C. Ng, Yue Lu, Inyoul Lee, Leroy Hood, Kai Wang

**Affiliations:** 1Institute for Systems Biology, Seattle, WA 98109, USA; suman.kumari@systemsbiology.org (S.K.); chris.lausted@systemsbiology.org (C.L.); kelsey.scherler@systemsbiology.org (K.S.); lee.hood@systemsbiology.org (L.H.); 2Department of Molecular Pharmaceutics, University of Utah, Salt Lake City, UT 84112, USA; alphonsus.ng@pharm.utah.edu (A.H.C.N.); yue.lu@pharm.utah.edu (Y.L.); 3Department of Biomedical Engineering, University of Utah, Salt Lake City, UT 84112, USA

**Keywords:** extracellular vesicles, EV biomarkers, EV RNA subtypes, RNA characterization, proteins characterization methods, single EV analysis

## Abstract

Extracellular vesicles (EVs) are lipid bilayer nanoparticles released from all known cells and are involved in cell-to-cell communication via their molecular content. EVs have been found in all tissues and body fluids, carrying a variety of biomolecules, including DNA, RNA, proteins, metabolites, and lipids, offering insights into cellular and pathophysiological conditions. Despite the emergence of EVs and their molecular contents as important biological indicators, it remains difficult to explore EV-mediated biological processes due to their small size and heterogeneity and the technical challenges in characterizing their molecular content. EV-associated small RNAs, especially microRNAs, have been extensively studied. However, other less characterized RNAs, including protein-coding mRNAs, long noncoding RNAs, circular RNAs, and tRNAs, have also been found in EVs. Furthermore, the EV-associated proteins can be used to distinguish different types of EVs. The spectrum of EV-associated RNAs, as well as proteins, may be associated with different pathophysiological conditions. Therefore, the ability to comprehensively characterize EVs’ molecular content is critical for understanding their biological function and potential applications in disease diagnosis. Here, we set out to provide an overview of EV-associated RNAs and proteins as well as approaches currently being used to characterize them.

## 1. Introduction

Extracellular vesicles (EVs) are heterogeneous lipid nanoparticles that are produced by all types of cells in the body [1,2]. They have been detected in various biological fluids, including blood, urine, saliva, follicular fluid, synovial fluid, and others [3,4,5,6,7,8,9]. EVs have been classified into different types based on their size, density, biochemical constituents, and biogenesis. They occur over a wide range of diameters, varying from 30 to 2000 nm, and contain nucleic acids, proteins, lipids, and metabolites that can reflect their cell of origin [10,11]. To standardize the nomenclature, the International Society for Extracellular Vesicles has developed the Minimal Information for Studies of Extracellular Vesicles (MISEV). This categorizes EVs based on size, with small EVs (sEVs) defined as those under 200 nm and large EVs (lEVs) over 200 nm, and defines types based on biogenesis, with exosome, small ectosome, microvesicle, and apoptotic vesicle subtypes arising from distinct processes [12]. Small particles released by cells that do not feature a lipid bilayer are not considered EVs.

EV subtypes produced by different mechanisms typically fall into different size ranges (Figure 1). Exosomes are a subtype of small extracellular vesicles, ranging from 30 to 150 nm, formed by the inward budding of the endosomal membrane [2,13]. On the other hand, small ectosomes, which have a similar size, are produced by outward budding from the plasma membrane [14]. Microvesicles, large extracellular vesicles, range from 100 to 1000 nm and are also formed by the outward budding of the plasma membrane [15,16]. Apoptotic vesicles are another class of larger vesicles, usually >1000 nm, and are produced from the plasma membrane by cells during apoptosis. They can encapsulate larger organelles and cellular content, which can be used as distinguishing features [17]. It is difficult to isolate pure forms of different types of EVs due to their overlapping sizes and physical characteristics, such as density. In terms of shape, exosomes and microvesicles are spherical, and apoptotic bodies are present in various shapes [18]. Moreover, EV heterogeneity has been observed in their biochemical content, reflecting different biogenesis processes and states of parental cells, which can impact experimental reproducibility [2,19,20]. EVs have been proven as valuable tools for biomarker discovery due to their ability to reflect the molecular characteristics of their parent cells [19,21]. The molecular content of EVs includes RNAs, proteins, metabolites, and lipids, which can be delivered and affect the activities of recipient cells [19,21,22,23]. EVs are increasingly recognized as significant tools in diagnosing and monitoring the therapy response of various diseases. Therefore, it is critical to develop methods that can accurately describe them to determine their origin and potential roles in physiological and pathological conditions. Factors that make isolation and characterization difficult include variations in physical properties and molecular contents, the dependence on the pathophysiological state of parental cells, and the lack of high-throughput methods to analyze individual EVs. Several techniques have recently been established to characterize the EV cargo, and more advanced approaches are continuously being developed for a better understanding of EVs. Our goal in preparing this review is to provide an overview of the two major biochemical cargos, RNA and protein, and their emerging significance in biomarker development, as well as the technologies employed to isolate and characterize EVs. We also highlight and discuss the benefits and drawbacks of various approaches, challenges in the field, and potential directions for future developments.

## 2. Key Methods for EV Isolation

Choosing an appropriate EV isolation method is crucial for preserving the structural integrity and specific functions of EVs, as well as for the subsequent characterization of molecular contents in subpopulations of interest. The MISEV2023 guidelines highlight the importance of evaluating different EV separation methods based on two key parameters: recovery and specificity [12]. This assessment not only provides a clear overview of the effectiveness of each method but also helps researchers in selecting the best approaches for their specific experimental goals. Several common methods with advantages and disadvantages for EV isolation are outlined in Table 1.

Ultracentrifugation (UC) is often referred to as the “gold standard” for EV isolation and concentration. While this method effectively separates sEVs from lEVs based on their sizes and densities, which helps reduce contamination, it may have limitations, such as potential aggregation and structural changes in EVs due to the high centrifugal forces [12,24,25,26]. A subsequent density gradient centrifugation (DGC) can enhance the purity of EV preparations obtained through UC, though it suffers from low throughput [27,28]. Recently, size exclusion chromatography (SEC) offers a gentler alternative to centrifugation, as it separates components based on their hydrodynamic volume, thus preserving the integrity of EV membranes. Despite this advantage, SEC is restricted by lower resolution, which may lead to the co-isolation of contaminants like viral particles and lipoproteins [29,30,31]. Field-flow fractionation (FFF) separates EVs and particles based on their size and molecular weight without using a stationary phase. These methods handle heterogeneous samples effectively and allow for high particle recovery [32,33]. Ultrafiltration is another prevalent method for isolating EVs, utilizing membranes with defined pore sizes to exclude larger particles. This technique is advantageous due to its simplicity, efficiency, and speed. However, it also has some limitations, such as lower yields compared to other isolation methods [34,35,36]. Recently, numerous commercial kits based on precipitation technology have been developed for EV isolation. The method is quick but also co-precipitates a significant amount of protein particles. In addition, chemicals such as polyethylene glycol used in the kit may hinder the downstream analysis [37,38]. Affinity-based isolation using antibodies targets specific surface antigens on EV, significantly increasing the purity of the isolated vesicles and speeding up the isolation process. These antibodies are typically immobilized on plates or magnetic beads; however, this method can be costly and may face challenges related to nonspecific binding [34,39,40].

To improve the purity and yield of EV, researchers often combine multiple isolation strategies. For example, utilizing UC alongside SEC or DGC can enhance the overall purity of EV preparations. Additionally, combining ultrafiltration with liquid chromatography has been effective in achieving higher yields. By integrating various isolation methods, researchers can optimize the quality and quantity of EVs for further analysis, ultimately enhancing the validity of their results [25,43]. Several innovative techniques have emerged for isolating EVs, particularly leveraging microfluidics, which aim to overcome the limitations associated with traditional methods. Microfluidic approaches facilitate the isolation of EVs from small volumes of biofluid samples, enabling more efficient, high-purity, and high-yield extractions [19,41,42]. A common method within these microfluidic systems is immunoaffinity capture, where microbeads coated with anti-CD63 antibodies serve as trapping arrays within a hydrodynamic device. This design significantly reduces background noise interference, thereby improving the purity of the isolated EVs [34,44].

However, research in this field faces challenges due to the lack of clarity on the classification of EVs and the effects of different isolation techniques on their molecular composition. This underscores the importance of comprehensively characterizing and isolating all EV subpopulations from any given source to enable a complete analysis of their molecular components [25,45]. If EV subtypes exhibit distinct functional roles, the use of impure samples could raise significant concerns in clinical applications. As the field continues to advance, it is essential to address the technical challenges associated with achieving homogeneous purification of EVs and to establish reliable quantification methods. This will ensure that the full potential of EVs is realized in both research and clinical settings.

## 3. RNAs in EV and Their Characterization

Various biological fluids such as saliva, plasma, serum, and urine contain cell-free RNAs, often referred to as extracellular RNAs [46]. The high blood RNase activities have led to the assumption that extracellular RNAs are fragile. However, studies have shown that extracellular RNAs, especially microRNAs (miRNAs), are very stable and even resistant to RNAse degradation and freeze–thaw cycles [47,48]. This is because extracellular RNAs are either encapsulated in EVs or associated with ribonucleoproteins and lipoproteins such as chylomicrons, very low-density lipoproteins, low-density lipoproteins, and high-density lipoproteins, which protect them from degradation [46,49,50]. Studies have demonstrated that RNAs, including miRNA, can be delivered in both EV and non-EV forms [49,51]. Since EVs are loaded with various bioactive cargos, such as nucleic acids (DNA and RNA), proteins, metabolites, and lipids that reflect the physiological state of originating cells, they allow the possibility to study normal and diseased tissues that are otherwise hard to access [19,46,52,53]. Furthermore, EVs contain a fair amount of RNA [54]. Therefore, it is useful to obtain a thorough profile of noncoding and protein-coding RNAs in EVs to understand their function as well as potential as molecular markers.

### 3.1. EV RNA Molecules

#### 3.1.1. Noncoding RNAs in EV

Noncoding RNAs (ncRNAs) may not be translated, but their transcription is crucial for shaping the cellular transcriptome and proteome. Based on the size, they are classified into small and long ncRNAs. The small ncRNAs include miRNAs and P-element-induced wimpy testis (PIWI)-interacting RNAs (piRNAs), which are around 18 to 29 nucleotides (nt), transfer RNAs (tRNAs) of 78 to 90 nt, and small nuclear RNAs (snRNAs) and nucleolar RNAs (snoRNAs) of 150 to 300 nt in length [55,56,57]. The long ncRNAs range from a few hundred (>200) to several thousand nt long [58]. The most abundant ncRNAs in cells are ribosomal RNAs (rRNAs), which may comprise about 80% of the total cellular RNA and are the predominant species in apoptotic bodies and large microvesicles but are less abundant in exosomes [59]. Nevertheless, the presence of various rRNAs in EVs poses some analytical issues.

Small ncRNAs, such as **miRNAs,** bind to the mRNAs in the cells, usually at the 3’ untranslated region (UTR), through partial sequence complementarity to affect the stability of mRNAs and protein translation [60]. MiRNA is one of the major components in human serum/plasma extracellular RNA [61]. Due to their significant role in biological processes and their presence in various biological fluids, miRNAs have become the most extensively investigated extracellular RNAs, particularly those associated with EVs [62]. A growing number of studies are characterizing extracellular miRNAs and suggesting them as non-invasive biomarkers because of their stability and potential role in the progression of a variety of diseases, including neurological disorders such as Alzheimer’s Disease (AD), Parkinson’s Disease (PD), traumatic brain injury (TBI), and post-traumatic stress disorder (PTSD). The differential expression of miR-212 and miR-132 in plasma EV of patients with AD may help in the early detection of AD [63]. Another work revealed that miR-29c-3p in plasma EVs is a promising diagnostic marker for AD [64]. Additionally, there is evidence showing the association between cerebrospinal fluid (CSF) EV miRNAs (miR-153, miR-409-3p, miR-10a-5p) and PD [65]. Following TBI, significant changes in miRNA profile have been observed in serum, plasma, and plasma EVs. Interestingly, some of these miRNAs are linked to the known TBI-related genes and pathways [66,67,68]. Several miRNAs from plasma EVs have shown a diagnostic potential for PTSD, including miR-203a-3p [69,70]. In addition to neurological disorders, the changes in EV miRNAs were found in metabolic disorders, including Type I diabetes (T1D) [71], hepatocellular carcinoma [72,73], non-alcoholic fatty liver disease [74], and diabetic nephropathy [75], which indicates the potential of EV miRNAs as non-invasive diagnostic markers for these diseases. 

**Small nuclear (SnRNAs) and nucleolar RNAs (SnoRNAs).** SnRNAs interact with small nuclear ribonucleoproteins to form the spliceosome complex and are involved in the splicing process during mRNA maturation [76]. SnRNAs can be detected through small RNA sequencing analyses and appear to be one of the major components in EV RNAs [77]. SnoRNAs participate in the modification and cleavage of rRNAs, tRNAs, and snRNAs [78]. Small RNA sequencing data showed their presence in EVs [79]. A recently published study showed that urinary EV snoRNAs can also hold great potential as biomarkers for cancers such as clear cell renal cell carcinoma [80]. The EV-encapsulated snoRNAs may also be involved in other biological activities. For example, Fitz and colleagues detected several specific snoRNAs enriched in plasma EVs of patients with AD compared to healthy controls [81].

**Circular RNAs. The circRNAs** are covalently closed RNA loops that are stable and extremely resistant to exonucleases. They are abundantly detected in the extracellular environment, including EVs, compared to tissue. They are known to act as miRNA sponges, blocking miRNA’s effects on downstream genes [82]. CircRNAs may also have other roles in normal or pathological processes since the progression of several neuropsychiatric disorders, including schizophrenia and depression, correlates with specific circRNA concentrations [83,84]. For instance, the concentrations of circRNAs, hsa_circ_0077837 and hsa_circ_0001495, from plasma EVs are significantly decreased in patients with schizophrenia compared to healthy controls [85].

**Transfer RNAs. The tRNAs** are highly structured molecules that serve as adaptor molecules in protein synthesis [86]. They have an amino acid attachment site and a 3-anticodon loop for recognizing mRNA. Recent studies suggest that tRNA-derived fragments may play some roles in immunomodulation and cancer development [87,88,89]. They are the most prevalent short noncoding RNAs in EVs, and the spectrum of tRNA fragments in EVs seems to be affected by the state of parental cells [87]. According to recent studies, tRNA fragments may be involved in stress signaling. The tRNA fragments have been observed in different body fluids and showed promise as disease markers for neurological diseases and cancers [90,91,92].

**P-element-induced wimpy testis-interacting RNAs (piRNAs).** Another class of small noncoding RNAs is the piRNAs. They are primarily involved in transcriptional and post-transcriptional gene regulation [93,94]. According to early estimates, piRNA makes up approximately 4% of the total RNA in human plasma EVs. Some recent reports indicate that piRNA levels are comparable to those of miRNAs [95]. The concentration of EV-containing piRNA correlates with a variety of cancers, including lung adenocarcinoma, non-small cell lung cancer, and prostate cancer in patients, compared to healthy controls, and these findings suggest their possible roles in cancer pathophysiology [96,97,98]. Additionally, EV-piRNAs exhibit good potential as prognostic markers in various diseases, though further work remains to establish clinical applications of EV-based piRNAs [99].

**Long ncRNAs. The lncRNAs** are transcripts larger than 200 nt without any protein-coding function [100]. They were once thought to be transcriptional noise because of their low concentration [101,102]. Compared to other non-coding RNAs, lncRNAs are poorly understood, and their diversity and widespread occurrence make them difficult to analyze and annotate. Generally, they can be divided into five categories based on their chromosomal location: (a) stand-alone ncRNAs, which do not overlap with protein-coding genes, (b) antisense transcripts, which have different levels of complementarity to sense transcripts, (c) pseudogenes, generated from genes that lost their coding ability due to mutation, (d) long intronic ncRNAs, produced during the transcription of protein-coding genes, and (e) long ncRNAs related to enhancers and promoters of protein-coding transcripts [101,103]. Several researchers have examined the enrichment of long ncRNAs in EVs and linked their possible involvements in biological processes such as angiogenesis, cell proliferation, apoptosis, and drug resistance in recipient cells [104,105,106]. For example, exosomal lncRNA, GAS5 (Growth Arrest Specific 5), produced from THP-1 cells, is known to be involved in vascular endothelial cell apoptosis [107]. Furthermore, recent studies have demonstrated the involvement of EV long ncRNAs in metabolic and neurological disorders, including diabetic retinopathy, type 2 diabetes (T2D), autism spectrum disorder, and PD [108,109,110,111]. For instance, the concentration of plasma EV lncRNAs, Linc-POU3F3 and lnc-MKRN2-42:1, correlate with the onset of PD [111,112]. Likewise, the level of long ncRNA, BACE1-AS, in plasma EVs is also increased in patients with AD [113]. These studies underline the usefulness of long ncRNAs in biomarker discovery. Despite these interesting findings, the overall biological activities for most of the long ncRNAs, especially those in EVs, remain undetermined.

#### 3.1.2. Protein Coding RNAs in EV

**Messenger RNAs.** Protein-coding mRNAs range in length from 150 to 60,000 nt and are also present in EVs. As most mRNA genes have known functions, they are of great interest for understanding EV biology and for possible clinical uses such as disease biomarkers and therapeutic targets [52]. Recently, a study conducted in maternal blood identified 181 female brain-specific mRNAs in blood EVs. These mRNAs are enriched in EVs that are associated with mood disorders, schizophrenia, and postpartum depression, highlighting their potential as biomarkers for mental health conditions [114]. Functional full-length mRNAs have been observed in EVs by several studies, whereas others have not been able to detect RNAs longer than 1000 nt in EVs [106]. In general, the length spectrum of EV mRNAs is shorter than that of cellular mRNAs. It has been observed that EV mRNAs derived from hepatocellular cancer cells were in the range of 50–4000 nt in length, whereas the corresponding cellular mRNAs were found in the range of 400–12,000 nt [106,115,116,117,118]. These findings may suggest that EVs contain both full-length and fragmented mRNAs. Some of the EV mRNAs may produce functional proteins and affect the function of recipient cells. For example, EVs from embryonic stem cells carry WNT3 mRNA, which can affect protein synthesis in target cells [119]. Similarly, Valadi and coworkers confirmed the existence of functional mRNAs in EVs generated from mast cells [23], offering the feasibility of the therapeutic utility of EV mRNAs. Furthermore, it has been proposed that CUEDC2 mRNA in the CSF exosome may serve as a potential biomarker for amyotrophic lateral sclerosis [120].

### 3.2. Methods to Characterize EV-Associated RNAs

EV-encapsulated extracellular RNAs have emerged as key signaling molecules, so the ability to characterize them will provide valuable insights into EV biology and provide potential biomarkers for different diseases. Therefore, it is imperative to develop reliable approaches to comprehensively characterize EV-associated RNAs. We outlined the major techniques currently used to characterize EV-encapsulated RNAs, including quantitative real-time PCR (qRT-PCR), microarrays, and RNA sequencing, and have summarized each technique with advantages and disadvantages in Table 2.

**qRT-PCR** has been a widely used approach to quantify specific RNA sequences such as EV miRNAs or mRNAs in biological samples [121,122,123,124]. The method requires two specific primers combined with thermostable DNA polymerase to amplify sequences of interest. This has been a standard method to assess the concentration of specific RNA molecules in biological samples due to its high sensitivity and low sample volume requirement [135]. Studies have demonstrated that qRT-PCR-based EV miRNA detection in the body fluids could be a potential diagnostic tool for several human diseases, including neurological and metabolic disorders, as summarized in Table 3. Even though qRT-PCR is rapid and reliable, some RNAs, such as miRNAs, present challenges because of their short lengths for standard PCR assays. Modifications have been made by either using polyA polymerase or hybridizing to a complementary sequence to extend the miRNA target sequence to develop qRT-PCR-based miRNA quantitation methods [136]. The other challenge for the miRNA qRT-PCR assay is the high degree of sequence similarity for some miRNA family members. Researchers are using primer overhangs or locked nucleic acid primers to increase the specificity [122,136]. With these modifications, the qRT-PCR assay can distinguish miRNAs with single nucleotide differences.

The use of proper control in quantifying specific transcripts in body fluid samples is another challenge. Glyceraldehyde-3-Phosphate Dehydrogenase (*GAPDH*), a ubiquitous gene, has been used as an internal reference to normalize qRT-PCR results in cells or tissue samples [137]. However, *GAPDH* is not found universally in the extracellular environment [138]. Small nuclear RNA U6 has been used to normalize the cell-free miRNA quantitation in different biological fluids, but reports have shown varying plasma U6 levels in different pathophysiological conditions [139]. Alternatively, spike-in synthetic non-human RNAs have been used for a more accurate normalization method in extracellular RNA studies [140].

Despite the high reliability and sensitivity of well-established assays, the qRT-PCR approaches have some limitations, including relatively low throughput and an inability to assess the distribution of miRNA length variations [141]. In addition, for the limited sample size and low abundance of extracellular RNA, when the concentration of target RNA is low, the result of qRT-PCR becomes unreliable since the measurement is nearing its detection limit. The newer **droplet digital polymerase chain reaction (ddPCR)** offers a higher sensitivity and more reliable method to measure the concentrations of low abundant transcripts. The ddPCR method is based on a limiting dilution principle, amplifying and counting individual target nucleic acids encapsulated in small droplets. The sample is divided into picolitre-sized droplets within an oil emulsion that serves as distinct PCR reaction chambers [125]. The concentration of the targeted molecule can be calculated based on counting the number of positive droplets. ddPCR provides better precision and reproducibility compared to traditional qRT-PCR. The method also allows the absolute quantification of low-abundance transcripts, such as some EV-associated transcripts [126]. In addition, the ddPCR eliminates the need for reference genes, which is a big advantage in measuring the concentration of specific RNA in the extracellular space. As ddPCR is based on the presence or absence of an amplification signal among individual droplets, the process is more tolerant to PCR inhibitors, which are common in various body fluids [125]. The newly developed multiplex ddPCR detects more targets than traditional ddPCR, providing more flexibility for extracellular DNA or RNA-based diagnostic applications. For example, a study conducted in 2023 used ddPCR to measure the concentrations of miR-223-3p and miR-7-1-5p in serum EVs to detect PD and demonstrated their use as reliable biomarkers [142].

**Microarrays** are a medium throughput platform that enables simultaneous quantification of thousands of RNAs, including miRNAs, using covalently attached nucleic acid probes on a solid support to hybridize with fluorescently tagged target sequences [127,128,129]. Researchers have adapted the microarray technique to profile the extracellular RNAs to discover biomarkers for brain and metabolic diseases such as multiple sclerosis, depression, schizophrenia, T1D, and T2D [143,144,145,146,147] (Table 3). In addition, the microarray-based technology was used to identify specific cell-free miRNAs involved in cystic fibrosis and acetaminophen-induced liver damage [129,148]. Microarrays have several limitations, such as lower sensitivity, smaller dynamic range, and can only detect known RNA sequences. Optimization is also required to avoid instances of cross-hybridization, as signals depend on the proper choice of probe sequences. It requires a substantial amount of RNA starting material, which can be a significant limitation for biological samples where EV yields may be low [127].

**Table 3 biomolecules-14-01599-t003:** Extracellular vesicle RNAs as biomarkers in brain and metabolic disorders.

Diseases		EVs Derivation	Isolation Method	EV RNA Biomarker and Alteration in Disease	Detection Method	Study
**Neurodegenerative Disorders**	**AD**	Plasma	Ultracentrifugation	miR-342-3p↓ miR-23b-3p↓	NGS	[149]
		Plasma	Polymer-Based Precipitation	miR-29c-3p↑	qRT-PCR	[64]
		CSF	Polymer-Based Precipitation	miR-212↓ miR-132↓	miScript miRNA PCR Array, qRT-PCR	[63]
		CSF	Ultracentrifugation	miR-153↑ miR-409-3p↑ miR-10a-5p↑	TaqMan Low-Density Array, qRT-PCR	[65]
	**PD**	Plasma	Polymer-Based Precipitation	let-7e-5p↑	NGS	[150]
		Plasma	Polymer-Based Precipitation	miR-30c-2-3p↑ miR-15b-5p↓ miR-138-5p↓ miR-338-3p↓ miR-106b-3p↓ miR-431-5p↓	NGS, qRT-PCR	[151]
		Plasma	Immunoprecipitation	lncRNA-Linc-POU3F3↑	Microarray, qRT-PCR	[111]
		Plasma	Ultracentrifugation	lncRNA- MKRN2-42:1↓	NGS, qRT-PCR	[112]
	**ALS**	Serum	Polymer-Based Precipitation	miR-27a-3p↓	qRT-PCR	[152]
		CSF	Centrifugation/Polymer-Based Precipitation	mRNA: CUEDC2↑	NGS, qRT-PCR	[120]
		Plasma	Polymer-Based Precipitation	miR-10b-5p↓ miR-199a-3p↑ miR-199a-5p↑	NGS, qRT-PCR	[153]
	**Multiple Sclerosis**	Plasma	Polyethylene Glycol Buffer/Sequential Centrifugation	miR-25↑ miR-19b↑ miR-92a↑ let7i↑	Microarray, qRT-PCR	[143]
		Serum	Polymer-Based Precipitation	miR-122-3p↓ miR-660-5p↑	qRT-PCR	[154]
		Plasma	Polymer-Based Precipitation	miR-150-5p↑ and let-7b-5p↓	qRT-PCR	[155]
		Serum	Polymer-Based Precipitation	miR-122-5p↓ miR-196b-5p↓ miR-301a-3p↓ miR-532-5p ↓	NGS, qRT-PCR	[156]
	**Depression**	Serum	Size Exclusion Chromatography	miR-139-5p↑	qRT-PCR	[157]
		Plasma	Polymer-Based Precipitation	miR-335-5p↑ hsa-miR-1292-3p↓	NGS	[158]
		Serum	Ultracentrifugation	miR-9-5p↑	Microarray	[144]
**Neuropsychiatric Disorders**	**SCZ**	Serum	Size Exclusion Chromatography	miR-206↑	NGS	[159]
		Plasma	Ultracentrifugation	miR-675-3p↑	Microarray, qRT-PCR	[145]
	**BD**	Plasma	Polymer-Based Precipitation	mir-484 ↓miR-652-3p↓ miR-142-3p↓ miR-185-5p↑	qRT-PCR	[160]
	**ASD**	Plasma	Polymer-Based Precipitation	lncRNAs-RP11-501J20.5↑ RP11-38L15.3↑ STX8↑	qRT-PCR	[110]
	**TBI**	Plasma	Ultracentrifugation	miR-1-3p↑ miR-143-3p↑ miR-151b↑ miR-27a-3p↑ miR-29a-3p↑	NGS	[67]
		Plasma	Polymer-Based Precipitation	miR-139-5p↓ miR-146a-5p↓ miR-21-5p↓ miR-483-5p↓	NGS, qRT-PCR	[68]
	**PTSD**	Plasma	Polymer-Based Precipitation	miR-139-5p↑	Ncounter Microarray	[70]
		Plasma	Size Exclusion Chromatography	miR-203a-3p↑	NGS, qRT-PCR	[69]
**Neural Tumor**	**GBM**	Serum	Ultracentrifugation	miR-210↑	qRT-PCR	[161]
		CSF	Ultracentrifugation	miR-21↑	qRT-PCR	[162]
		Serum	Polymer-Based Precipitation	miR-454–3p↑	qRT-PCR	[163]
**Metabolic Disorders**	**T2D**	Serum	Polymer-Based Precipitation	lncRNA-p3134↑	Microarray	[109]
		Serum	Polymer-Based Precipitation	miR-20b-5p↑	qRT-PCR, Microarray	[147]
	**DN**	Urine	Ultracentrifugation	miR-130a↑ miR-145↑ miR-155 ↓miR-424↓	qRT-PCR	[75]
	**DR**	Plasma	Size Exclusion Chromatography	miR-15a↑	qRT-PCR	[164]
		Plasma	Polymer-Based Precipitation	lncRNAs: DLX6-AS1 ↑ PRINS↓ FAM190A-3↓	NGS, qRT-PCR	[108]
	**T1D**	Serum	Sequential Ultracentrifugation/Polymer-Based Precipitation	miR-21-5p↑	qRT-PCR	[71]
		Plasma	Differential Ultracentrifugation	miR-16↓ miR-302d-3p↓ miR-378e↓ miR-570–3p↓ miR-574-5p↓ miR-579↓ miR-25-3p↑	Microarray, qRT-PCR	[146]
	**NAFLD**	Plasma	Size Exclusion Chromatography	miR-128-3p↓	qRT-PCR	[74]
		Serum	Polymer-Based Precipitation	miR35a-3p↓ miR-129b-5p↓ miR-504-3p↓ miR-122-5p↑	qRT-PCR	[165]
	**HC**	Serum	Polymer-Based Precipitation	miR-224↑	qRT-PCR	[73]
		Serum	Polymer-Based Precipitation	miR-320d↓	qRT-PCR	[72]

Abbreviations: AD, Alzheimer’s Disease; PD, Parkinson’s Disease; ALS, Amyotrophic Lateral Sclerosis; SCZ, Schizophrenia; ASD, Autism Spectrum Disorder; TBI, traumatic brain injury; PTSD, post-traumatic stress disorder; GBM, glioblastoma; T2D, Type 2 Diabetes; DN, diabetes nephropathy; DR, Diabetic Retinopathy; T1D, Type 1 Diabetes; NAFLD, non-alcoholic fatty liver disease; HC, hepatocellular cancer; CSF, cerebrospinal fluid; lncRNA, Long noncoding RNA; NGS, next-generation sequencing; ↑, increased abundance; ↓, decreased abundance.

**Next-generation Sequencing (NGS)**. The increased dynamic range, higher throughput, and target identification accuracy make RNA sequencing the method of choice to characterize extracellular RNA. Unlike microarrays or qRT-PCR, the NGS platform does not need to use transcript-specific probes or primers and allows the identification of novel transcripts and variants, such as gene fusions and small insertions and deletions [130,131]. However, the low typical concentrations of extracellular RNA make it technically difficult to obtain a comprehensive profile of RNA. Various processes like rRNA depletion have been used to improve the detection of less abundant RNA types (noncoding RNAs, snRNAs, snoRNAs, piRNAs, tRNAs) in EVs [59,166,167]. In addition, it has been shown that removing rRNAs from plasma RNA improves the fraction of reads mapped to RNA of interest [168]. However, the NGS approach is more vulnerable to sequence biases, especially for small RNA sequencing (sRNAseq). For instance, it has been noted that significant biases were added during the ligation step in small RNA sequencing library preparation. Different sRNAseq library preparation methods produce noticeably different sequencing outcomes [132,133,134]. It was recommended to use degenerate adapters in miRNA library construction to reduce the bias caused by ligases and sequence composition. Additionally, it was recommended to use synthetic spike-ins to control RNA isolation and library construction to ensure the accuracy of small RNA profiling [169]. Most sequencing platforms, such as Illumina’s sequencing-by-synthesis system, require expensive instrumentation, complex data processing, and relatively long turn-around times [132]. Nevertheless, this approach is continually evolving and is currently the gold standard for characterizing cell-free RNAs. Successful applications include the study of neurodegenerative and neuropsychiatric diseases, including AD, PD, ALS, multiple sclerosis, depression, and schizophrenia (See Table 3) [149,151,153,156,158,159].

## 4. Protein Molecules in EVs and Their Characterization

Protein is the fundamental execution unit of most biological processes. Proteins found on the surface and inside of EVs may reflect the cell types from which they originated and their biological functions; therefore, they are important for understanding EV biology as well as for biomarker discovery [170]. Determining the overall protein profile is a vital step toward understanding the function and heterogeneity of EVs. Different types of EVs usually carry some unique proteins, which can be utilized to classify EVs. For example, the exosomes secreted from cells are known to be mediated by tetraspanin surface proteins such as CD9, as well as ALIX, flotillin, and TSG101 [53,171]. Some of these proteins remain on the exosome surface and can be used to capture and enrich exosomes. The CD9-positive exosomes can carry very different protein cargo, which might be due to the different types and states of the parental cells [53,171,172]. Exosomes also contain cytoskeletal proteins such as actin, myosin, and heat shock proteins like Hsp70 [26,173]. Microvesicles, on the other hand, typically include CD40, integrins, glycoproteins, and P-selectin [174]. They may involve specific functions of the cells. For example, ARF6 (ADP-ribosylation factor 6), a small GTP binding protein present in tumor-derived microvesicles, is probably involved in the secretion of proteases of tumor cells [175]. Apoptotic bodies contain cytoplasmic as well as nuclear proteins. In general, the protein cargo in apoptotic bodies is more complex than other types of EVs. Membrane proteins on apoptotic bodies include calreticulin and calnexin, which promote efferocytosis and the formation of phagolysosomes [176,177].

There are several online databases like Vesiclepedia (http://microvesicles.org/, accessed on 10 June 2024), EVpedia (https://evpedia.info/evpedia2_xe/, accessed on 10 June 2024), and ExoCarta (http://www.exocarta.org/, accessed on 12 June 2024) cataloging EV cargos, including proteins from different species. Both Vesiclepedia and EVpedia contain data from different types of EVs, while ExoCarta mainly focuses on exosome cargo [178]. Currently, Vesiclepedia has a total of about 566,911 protein entries, whereas EVpedia has about 558,045 protein entries [179,180]. About 117,606 exosome protein entries have been reported in ExoCarta [178]. The identification of EV-associated proteins, both within and on the surface, may aid in our understanding of the molecular processes underlying EV biogenesis, their origin, and their functions. It may also advance the development of EV-based biomarkers for diseases.

The total protein concentration in EVs can be determined using common colorimetric assays such as the **Bradford assay and micro-bicinchoninic acid** (**BCA**) **assay** [26]. Although these assays are fairly accurate and have been used routinely in the lab, the measurement may be affected by the purity of EVs. For example, high molecular weight proteins or protein complexes are often co-purified with EVs, which may impact the accuracy of EV total protein concentration measurement [181,182]. Several targeted or global experimental approaches have been used to assess the concentration of specific proteins or the overall spectrum of EV-associated proteins. These include traditional methods such as Western blot, ELISA, and mass spectrometry (MS) (Table 4). However, these methods usually require higher protein concentrations with significant sample volumes, posing challenges for studying EV proteins in human samples. Currently, single-particle analysis methods have been developed to characterize EVs’ biochemical and genetic content as they provide some advantages over the traditional methods.

### Methods to Characterize EV-Associated Proteins

**Western Blotting** (**WB**) is a method to detect specific proteins in the sample and is a tool researchers routinely use to assess the purity of EV. For example, the International Society for Extracellular Vesicles (ISEV) recommends characterizing multiple proteins, including CD9, CD63, CD81, TSG101, and ALIX. Quantification of protein contaminants, such as apolipoproteins A1/2, albumin, and uromodulin, is also recommended to evaluate EV [185,214]. To detect EV proteins by WB, purified EVs must first be lysed to release their protein contents. The lysate is then either directly spotted onto a membrane for dot blot analysis or separated by size and/or charge through a gel matrix such as sodium dodecyl sulfate–polyacrylamide gel electrophoresis (SDS-PAGE) [181,183]. The proteins of interest are then detected using antibodies specific to the proteins. WB is widely used to validate the EV protein biomarkers in various diseases, such as PD, schizophrenia, ASD, ALS, and certain metabolic disorders (e.g., T1D and T2D), as summarized in Table 5 [215,216,217,218,219,220]. WB is the most commonly used technique for analyzing EVs due to its simplicity, accessibility, and ability to detect both surface and internal proteins. However, it is a semi-quantitative method and has some limitations. A recent study discussed difficulties encountered while performing WB for EV markers CD9 and CD81. The overlapping molecular weights of these markers complicated the re-probing of PVDF membranes. Additionally, the small sample volumes available restricted the ability to run separate gels. The presence of a high background further affected the accuracy of densitometric measurements [184]. Moreover, it cannot reveal how much of a specific protein is in individual EVs or the degree of population heterogeneity among EVs; it only provides information on overall protein concentration in isolated EVs. Additionally, EVs lack a common reference protein for sample normalization in WB experiments, so equal protein concentration, sample volume, or particle number are commonly used. Considering that EV isolation is often time-consuming and produces low yields, multiplexed analysis methods are valuable for improving throughput by detecting multiple biomarkers at once. Different isolation methods selectively enrich specific EV sub-populations and vary in terms of recovery and contamination levels. Therefore, it is crucial to ensure compatibility between the isolation method and the characterization approach to optimize EV protein detection. Combining multiplexed detection with optimized isolation techniques can enhance yield and provide more reliable data for biomarker discovery and clinical applications [181,185,221].

**Enzyme-Linked Immunosorbent Assay** (**ELISA**) is frequently employed for the detection and measurement of specific proteins. Like WB, ELISA can be used to detect EV-associated proteins, including tetraspanin proteins (CD9, CD63, CD81) that are usually found on the exosome surface [186,249,250]. ELISAs are already being used in EV biomarker research [70,219,225,228,229,234,237,238], offer better sensitivity and accuracy, and require lower sample volumes when compared to WB (Table 5). ELISA is a multistep assay, which may cause some potential problems, such as high background, poor replication, and weak signal. ELISA still needs significant amounts of EVs, making it challenging to characterize EVs from human biological samples [185,187]. While ELISA is useful in evaluating the concentrations of specific proteins, it can be time-consuming and prone to technical errors due to variations associated with EV isolation and multiple experimental steps [187,188].

Fluorophore-linked immunosorbent assay (FLISA) and time-resolved-fluorescence immunoassay (TRFIA) are alternatives to ELISA that employ fluorescent conjugated antibodies rather than HRP antibodies to detect capture EVs. They have been utilized to detect EV surface proteins in blood and urine without requiring EV isolation, providing enhanced sensitivity [186,251,252]. For instance, Duijvesz and colleagues developed a highly sensitive TRFIA to detect prostate cancer-derived exosomes. They validated the techniques with urine samples, using biotinylated antibodies against CD9 or CD63 immobilized on streptavidin-coated wells, along with Europium-labeled detection antibodies. With its high sensitivity and minimal background noise, the TRFIA has proven to be an efficient technique for effectively measuring elevated CD9/CD63 levels in patients with prostate cancer. This assay serves as a foundation for developing disease-specific exosome detection tools [252]. 

**Flow cytometry** (**FC**) is another EV characterization technique adapted from the analysis of cells [253,254]. However, conventional FC has limited capability in analyzing particles smaller than 200 nm due to their detection sensitivity and lower refractive index [132,255,256]. For example, the ability of EVs to scatter laser light is more than ten times less compared to polystyrene beads, which are generally used for FC calibration [190]. To address this, fluorescent antibodies have been used to tag EV surface proteins, which allows the detection and quantification of EVs through FC in the 100 nm size range [257]. Besides tagging the EVs with fluorescent antibodies, particles smaller than 200 nm can also be detected with a flow cytometer using improved detectors and optimized laser excitation energy [191]. Additionally, the development of a commercial imaging flow cytometer (IFC), which combines conventional flow cytometry with fluorescence imaging capability, has made it possible to separate EVs from protein aggregates [189,192]. IFC includes a charge-coupled device with greater dynamic range and less noise than a photomultiplier tube and is better suited for monitoring 100–200 nm EVs which typically exhibit low fluorescence signals [192]. For instance, Ricklefs and colleagues utilized IFC to characterize multiple proteins on the EV surface, demonstrating the presence of EV subpopulations based on specific tetraspanin profiles. They found that CD63+/CD81+ double positive EVs were enriched in cancer cell lines and plasma from patients with malignant brain tumors [193]. 

Furthermore, the development of a compact Nanoscale Flow Cytometer (nFC) with multiplex fluorescent probes for exosomes and microvesicles also offers a more precise analysis of EVs, allowing detection of particles as small as 40 nm [132,258]. Recent research also highlights the importance of sample dilution in preventing swarm detection, where overlapping signals from multiple EVs in the illuminated region are mistaken for a single event. This method ensures accurate individual EV detection and minimizes measurement errors [259,260]. Advancements in the FC field now allow researchers to profile multiple proteins on individual EVs, enabling the determination of vesicle heterogeneity, their possible cellular origin, and their association with specific pathophysiologies [236,261]. Additionally, this technique offers the advantage of rapidly analyzing minimal biological samples, delivering reproducible and precise quantitative data without requiring EV isolation. While it serves as a valuable tool for studying EVs as potential clinical biomarkers [132,195], its reliance on specialized equipment remains a notable limitation, restricting its broader use and accessibility in clinical applications.

Over the past few years, comprehensive EV proteome analysis has been made possible by using MS-based methods. **MS** can identify and detect a wide variety of EV proteins in normal and diseased conditions and can lead to the discovery of biomarkers. For instance, Cai and coworkers used MS-based proteomics to profile the plasma exosomal proteome in patients with AD and reported that plasma exosomal proteins (A0A0G2JRQ6, C1QC, CO9, GP1BB, and RSU1) act as a novel candidate biomarker to differentiate patients with AD from healthy individuals. Additionally, the abundance of AACT (Alpha-1-Antichymotrypsin) in EVs from the serum of patients with AD patients also correlates with disease status. These findings imply the possible use of specific EV proteins as a disease biomarker, which could transform the screening and monitoring of diseases [223,224]. Furthermore, as demonstrated in Table 5, exosomal protein biomarkers were found via MS-based profiling methods in several diseases.

**Liquid chromatography–electrospray ionization tandem mass spectrometry** (**LC-ESI-MS/MS**) has been used to examine the molecular composition of EVs not only in neurodegenerative disorders but also in metabolic disorders. The identity of proteins in the samples can be provided by fragment ions from ESI-MS/MS [196]. To improve sensitivity, researchers have combined the ESI-MS/MS with HPLC, UPLC, and Nano-LC to profile proteins in EVs from various biological fluids. These methods facilitate the identification and quantification of both EV and non-EV proteins, making them powerful tools for comprehensive EV proteomic characterization. Particularly, nano-ESI–MS/MS stands out for its ability to detect and characterize vast numbers of proteins from EV samples with remarkable sensitivity and resolution [196,197]. This technology has become indispensable for detailed EV protein identification and composition profiling in various biological fluids, especially when compared to traditional methods like WB and ELISA [198,199,200]. A recently published study demonstrated that LC-MS/MS offers superior sensitivity for EV analysis, achieving picomolar detection levels with only 12.5 µL of the sample compared to the ∼100 µL required for WB. This method’s high sensitivity, antibody independence, and multiplexed quantification provide deeper insights into the protein content of EVs [184]. Although MS-based methods have improved our understanding of protein content in EVs, they still usually require multiple pre-processing steps, such as enzymatic digestion (usually trypsin) and peptide separation, before MS analysis [202]. The biggest challenge in current MS-based EV proteomics is the lack of robust and efficient methods for rapid and reproducible EV isolation [201].

**SOMAscan,** offered by Somalogic, enables the analysis of up to 11,000 proteins per sample simultaneously using DNA-based affinity agents called SOMAmers. SOMAmers are chemically modified nucleotide aptamers that bind to target analytes with a slow off rate [203,204]. Since aptamers have excellent stability, sensitivity, and a broader dynamic range (up to 10 logs), the SOMAscan is an attractive technique for processing a large number of samples, even when the samples are in limited quantities and can detect less abundant proteins. Webber et al. used the SOMAscan to study exosomes from patients with prostate cancer and discovered several novel prostate cancer-associated proteins, which may lead to the development of new cancer biomarkers [262]. Cross-reactivity and non-specific binding, however, can compromise the SOMAscan-based protein profiling [205,206]. Additionally, there is a higher likelihood of missing signals from low-abundant proteins. Low protein concentrations from EVs may lead to reduced specificity and significant intensity fluctuations between samples. While the high multiplexing approach like SOMAscan is useful for profiling EV-associated proteins, there is a need for more sensitive techniques to detect low abundant EV proteins.

Another multiplex protein detection approach is based on a **proximity ligation assay** (**PLA**) **or proximity extension assay** (**PEA**) that allows the simultaneous detection of thousands of proteins in samples, including EVs. A key advantage of these methods is the need for a very small amount of starting material, which does not require sample pre-processing before analysis. The PLA and PEA methods work on a similar principle, using matched pairs of DNA oligonucleotides and pairs of antibodies to analyze the proteins in biological samples, including EVs and biological fluids [207]. When pairs of antibodies bind to target antigens, the tagged oligonucleotides come into proximity. In PEA, they can form a short double-stranded DNA, which is then extended through DNA polymerization and quantified via qPCR. In PLA, the oligonucleotides hybridize with bridge oligos, followed by ligation with ligase and signal amplification through rolling circle amplifications [207,263]. The commercial version of PEA provided by Olink (Uppsala, Sweden) can analyze several hundreds to thousands of proteins with a small amount of sample [264,265]. For example, it can detect and quantify 92 different proteins from just 1 μL of sample such as serum or plasma [265,266]. One significant advantage is that the method does not need sample pre-processing, as required by MS. Recently, the PCR-based readout for PEA was changed to NGS, which increased its dynamic range and the number of proteins that can be analyzed. The most recent PEA assay from Olink allows the quantification of more than 5400 proteins, easily assembled from a group of 384-plex panels [208,209,210]. Overall, it was found that PEA offers good sensitivity, accuracy, and specificity in profiling proteins from biological samples, including EVs and different types of body fluids [68,264,267]. PEA has been employed in biomarker discovery, identifying potential EV-based protein biomarkers for conditions such as chronic mild TBI, sports-related brain trauma, AD, myocardial infarction, T2D, and glioblastoma [68,268,269,270,271,272]. Compared to other assays, PEA has several advantages, such as quick and simple experimental protocol, no sample pre-processing, multiplexing capability, and minimal sample consumption. We anticipate that this technology will be useful in biomarker research, especially in the analysis of low-abundant proteins, as well as for clinical samples with limited volume.

## 5. Recent Developments in Single EV Analysis Can Improve the Biomarker Detection

Even though it is clear that EVs have great potential in biomarker discovery for various diseases, there are still many obstacles to overcome. One major challenge is that EVs produced by individual cells exhibit significant heterogeneity and contain very few molecular markers in nanosized particles, making them difficult to study at the single-EV level with current methods. Characterizing the biochemical content of EVs at a single-EV resolution not only addressed the heterogeneity issue but also provided a detailed understanding of the origin and potential function. Particularly, it may allow the distinction between EVs produced by diseased and healthy cells, which would be beneficial for early diagnosis of diseases. Simultaneous detection of several EV constituents, such as RNAs and proteins, will make EV analysis more valuable and informative, particularly in the early stages of diseases [273].

**Microfluidics** has advanced single-cell research by offering a variety of methods for gathering transcriptome and proteome data from a large number of individual cells. Microfluidic devices can separate cells in picoliter droplets, and with proper cell concentration, each droplet may encapsulate just one cell, enabling detailed single-cell analysis [274]. However, adapting this approach to study EVs presents several challenges. For example, exosomes are typically about a million times smaller than cells in terms of mass or volume, and their endogenous RNAs and proteins are present in much lower quantities compared to those in cells [275]. Recently, several novel analytical methods have been developed in an attempt to characterize the molecular content of individual EVs. 

**Immuno Droplet Digital PCR** (**iddPCR**) is an ultrasensitive antibody-based method developed to detect specific proteins in single EVs. Briefly, antibody–DNA conjugates were used to target specific EV proteins, and then individual EVs were encapsulated in the droplets using microfluidics. PCR is then used to amplify DNA barcodes on the antibodies for analysis [276,277]. Recently, Liu et al. developed a similar method to group EVs in different subpopulations based on the composition of surface proteins on individual EVs. They found that EV subpopulations from a hepatocellular carcinoma cohort differed from those of healthy controls [278]. Therefore, this approach offers an interesting strategy to detect specific diseases based on the profile of changes in EV subpopulations. The benefits of this method include the ability to detect rare proteins of interest, a high signal-to-noise ratio, and excellent sensitivity [277]. However, this approach is limited by its capacity to detect only three proteins simultaneously due to its reliance on three fluorescent color channels. Additionally, the method cannot provide absolute protein levels in EVs because of the PCR-based readout [277].

Recently, a **single EV immune sequencing** (**seiSEQ**) technique has been developed. The method is based on a single-cell protein profiling approach. Individual EVs were tagged with DNA-conjugated antibodies and encapsulated into droplets containing barcoded beads using a microfluidic device [275]. Based on the unique DNA sequences from the barcode and antibody–DNA conjugates, the protein profile of individual EVs can be obtained through sequencing. The seiSEQ method offers a robust, sensitive sequencing-based readout with scalable multiplexing capabilities and can profile a large number of EVs. This approach provides an informative tool to characterize rare and distinct subpopulations of EVs [279] and can be used to analyze rare proteins in EVs. It has the potential to provide insights into EV biology and aid in the development of EV-based biomarkers.

**Proximity barcoding assay** (**PBA**) is a method developed for profiling surface proteins on individual EVs using antibody–DNA conjugates with NGS. In this method, exosomes are first tagged with antibodies conjugated to oligonucleotide barcodes, each containing a protein tag to barcode the target protein and a unique molecular identifier (UMI) to distinguish individual molecules. The exosomes are then captured in microtiter wells coated with cholera toxin subunit B, which binds to GM1 gangliosides on the exosome membrane. Separately, circular DNA oligonucleotides with unique random sequences, called complexTags, undergo rolling circle amplification (RCA), producing RCA products containing many copies of the same unique sequence. These RCA products are added to the wells, where each exosome associates with one RCA product, tagging the antibody–oligonucleotides with the unique complexTag during DNA polymerization. Subsequently, through PCR, each antibody generates multiple copies of a cDNA library element containing the protein tag, UMI, and complexTag. This allows sequencing to identify which proteins originate from the same exosome and enables the quantification of protein molecules per exosome. Using this approach, 207 surface proteins on exosomes, including key markers like CD9, CD63, and CD81, were successfully profiled [211]. A recent study applied PBA to explore the potential of EV proteins as biomarkers for cancer liquid biopsy [212]. Another study used PBA to characterize EV surface proteins in five distinct body fluids, investigating the non-invasive diagnostic markers for AD, with urinary EV proteins identified as the most promising markers [213]. Thus, based on differences in surface protein composition, PBA can be used to distinguish between different exosome populations that may be released into different body fluids from particular tissues in both healthy and diseased states.

## 6. Conclusions

EVs are small lipid bilayer nanoparticles that have gained interest for their biological functions and potential clinical applications in therapeutic development and disease diagnostics. However, the heterogeneity of EVs in biofluids, including variations in morphology, size, molecular content, biogenesis, and cell sources, makes it difficult to decipher their biological functions and develop their applications [280]. This heterogeneity also presents challenges in interpreting the function of EVs and detecting low-abundance biomolecules using bulk techniques such as MS, SOMAscan, Olink, WB, ELISA, and PCR. Bulk analytic approaches provide only the average characteristics of all EVs, significantly hampering their clinical applications. Emerging techniques such as PBA, nFC, and seiSEQ can now pinpoint details such as size and molecular makeup at the individual vesicle level. To fully realize the potential of EVs in clinical use, it is essential to characterize individual EVs to gain insight into their molecular cargo and potential biological roles.

Accurate interpretation of EV-related studies must consider the analytical approaches used, including whether they involve bulk or single-EV analysis, the types of EV studied, and the methods of EV isolation. Establishing standards in EV analysis methods, such as sample preparation, EV isolation, and data analysis, is critical to further advancing EV-based research. Even though single EV characterization approaches have significantly improved our understanding of the molecular markers of EVs, more work is still needed to develop platforms that offer higher multiplexity, lower experimental complexity, and greater affordability. Moreover, an even bigger challenge is translating the findings from individual EVs into biological insights and clinical applications.

A streamlined, integrated approach that allows for fast and reproducible EV isolation and analysis could open new avenues for EV-based applications. With advances in technologies such as microfluidics and lab-on-a-chip, there is a growing potential to combine EV isolation, purification, and molecular analysis within a single process. These integrated platforms could also enable the characterization of EVs from small volumes of clinical samples. Further developments in this area will likely lead to technologies that provide low-cost, rapid, and accurate EV quantification and characterization for clinical use. As the field continues to evolve, these advancements hold the promise of deepening our understanding of EV biology, unlocking new biomarkers, and driving innovative approaches for disease monitoring and precision medicine.

## Figures and Tables

**Figure 1 biomolecules-14-01599-f001:**
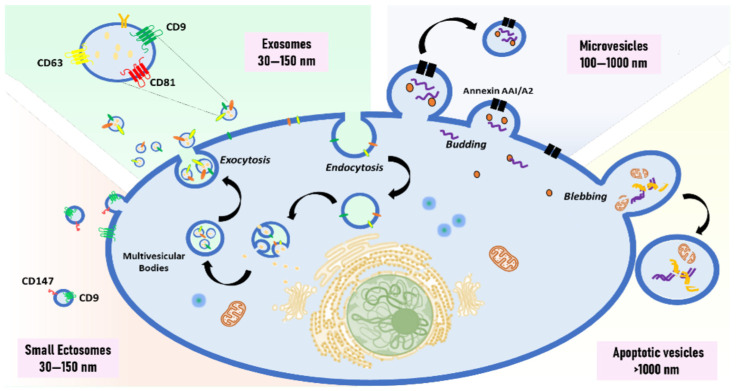
Depiction of EV subtypes and their biogenesis. Exosomes, a type of small EVs, are formed from the invagination of the endosomal membrane. Small ectosomes, another type of sEVs, are generated by the outward budding of the membrane. Microvesicles, indicated as large EVs, are produced through the direct budding of the plasma membrane. Apoptotic vesicles, a type of lEV, are produced by blebbing and fragmentation during apoptosis.

**Table 1 biomolecules-14-01599-t001:** Potential advantages and disadvantages of different EV isolation methods.

Methods	Advantages	Disadvantages	Study
**Ultracentrifugation**	Separation of EVs based on size and density Adaptability: method can be adjusted to target either lEVs or sEVs Standardized protocol	Incomplete Separation Low yield for sEVs High speed can cause aggregation and fusion of EVs Resource intensive	[12,24,25,26]
**Density Gradient Centrifugation**	Effectively separate EVs from non-vesicular extracellular proteins Size based fractionation	Optimal separation requires lengthy UC Low recovery rates Medium removal required Time consuming Careful handling of fractions is necessary to avoid disrupting the gradient	[27,28]
**Size Exclusion Chromatography**	Size based separation Enable to preserve the biological properties of EVs Can combined with other methods to enhance the separation of EVs Some SEC can be thoroughly cleaned and reused which reduce costs and wastes	Cause dilution of samples Abundance and purity of EVs in collected fractions must be characterized Variables such as flow rate and sample concentration must be carefully controlled to optimize separation	[29,30,31]
**Field-Flow Fractionation**	Allow better recovery of EVs Effective for heterogenous samples Gentle separation method	Not specifically designed for EVs Complex methodology Requires expensive instrument	[32,33]
**Ultrafiltration**	Higher yield Can be scaled up or down Ease of use Relatively low cost	Loss of EVs Chances of membrane blockage EV adherence Variable recovery	[34,35,36]
**Precipitation**	Simple and fast High yield Commercially available kits	Not ideal for high purity needs Can coprecipitate non-EV particles Lacking transparency in chemicals used in isolation media Coprecipitation of large amount of proteins	[37,38]
**Affinity Based Isolation**	Enable to yield highly pure EVs Potential for high throughput Can target various EV subtypes Noninvasive method	Affinity reagents such as antibodies or aptamers are often expensive Batch to batch variation in affinity reagents Difficulty in eluting EV particles Affinity reagents or matrix components may remain bound to EVs which can affect downstream analysis	[34,39,40]
**Microfluidic Approaches**	Rapid processing of small sample volumes Enhanced purity Integration with other technologies	Design and optimization of microfluidic devices can be technically challenging High initial setup cost may limit accessibility for some laboratories	[19,41,42]

**Table 2 biomolecules-14-01599-t002:** Advantages and disadvantages of different EV RNA characterization methods.

Technique	Approach	Application	Advantages	Disadvantages	Study
**Polymerase Chain Reaction**	Quantitative real-time PCR	mRNAs, long and small noncoding RNAs	Using standard PCR reagents qPCR is relatively quick Can identify differentially expressed RNA sequences in EVs	Needs reference transcript for normalization Primer design can affect the accuracy of measurement Sequence composition may affect the amplification Can measure only known transcripts	[121,122,123,124]
	Droplet digital PCR	miRNA, mRNAs, long and small noncoding RNAs	Provide absolute quantification Better sensitivity and accuracy, particularly for low concentration targets More tolerant to PCR inhibitors compared to qRT-PCR	Experimental optimization is needed Need specialized equipment Can only measure known transcripts	[125,126]
**Microarray**	Microarray	miRNA, mRNAs, long and small noncoding RNAs	Provide comprehensive profiling of RNAs in EVs Customized arrays are available for different types of RNA Standard informative tool for analysis High throughput and less expensive	Require a substantial amount of starting material Can measure only known transcripts Low sensitivity	[127,128,129]
**Next Generation Sequencing**	Short Read Next Generation Sequencing	miRNA, mRNAs, long and small noncoding RNAs	Provide comprehensive profiling of EV transcriptome Ability of high throughput identification and quantification of novel RNA transcripts in EVs Unique molecular identifiers improves the detection of low abundant sequences High throughput	Significant variations associated with small RNA sequencing library construction Need special reagent and equipment Library preparation process can introduce biases Challenging for low yield samples like EVs from clinical samples Need informatic support	[130,131,132,133,134]

**Table 4 biomolecules-14-01599-t004:** Advantages and disadvantages of different EV protein characterization methods.

Methods	Processing Time	Advantages	Disadvantages	Study
**Micro-Bicinchoninic Acid Assay**	2–3 h	Rapid and inexpensive method for protein quantification	Large sample volume required Not able to obtain more information from specific proteins	[26,181,182]
**Western Blotting**	>10 h	Well-established protocols Simplicity and availability Able to analyze EV purity by detecting EV positive and EV negative markers Can verify the molecular weight of EV proteins	Results are influenced by the quality of antibodies Limited multiplexing High sample/protein requirement Unable to analyze the proteins in intact vesicles Lengthy processing time	[181,183,184]
**Enzyme-Linked Immunosorbent Assay**	Several hours	Less expensive and easy to perform High specificity and sensitivity Reduced instrumentation	Error prone approach due to multistep procedure Quality of antibodies may affect the results	[185,186,187,188]
**Flow Cytometry**	45 min to 1 hour	Can subtype EVs based on the presence of different proteins High throughput Possible to assess specific markers at single EV level	Difficult to characterize smaller EVs EV-associated proteins might not be able to detectable all the time Require optimization of sample preparation and a skilled operator and numerous optimizations	[189,190,191,192,193,194,195]
**Mass spectrometry**	Days	Comprehensive proteomic profiling Provide quantitative information for target proteins High throughput and high sensitivity for EV proteins Enable amino acid composition and precise structural information of the EV proteins Reduced sample consumption compared to WB	Costly equipment Complex sample pre-processing Need informatic support Labor-intensive procedure	[184,196,197,198,199,200,201,202]
**SOMAscan**	<11 h	Simultaneous analysis of up to 11,000 proteins Need only small amount samples Reduces sample preparation complexity for challenging EV samples Large dynamic range Novel protein identification Partially automated	Low specificity and high cross reactivity due to unidentified epitopes High probability of false positive signals Need informatic support Array dependent Bias	[203,204,205,206]
**Proximity Ligation Assay,** **Proximity Extension Assay**	2 days	Simultaneous analysis of several thousand proteins Need very small amount samples Excellent detection of low abundant proteins Olink Target panels enable targeted analysis of proteins in plasma and lysed EVs Do not need sample pre-processing and is partially automated	Depend on the availability and quality of the antibody Covalent conjugation of oligonucleotides to antibodies can be challenging and time-consuming Selection of proximity probes that are ideal for the target antibodies is crucial Absolute quantification of target protein cannot be achieved Need informatic support	[207,208,209,210]
**Proximity Barcoding Assay**	2 Days	Simultaneous analysis of several proteins on individual EVs Highly sensitive Large dynamic range Exosome can be distinguished by their different surface protein composition Do not need sample pre-processing and special equipment	Depend on the protein combination specificity and antibody affinity Limited by the 207-protein panel, potentially restricting detection of exosomes from specific tissues May require tissue-specific proteins to improve exosome detection Absolute quantification of target protein cannot be achieved Need informatic support	[211,212,213]

**Table 5 biomolecules-14-01599-t005:** Extracellular vesicle proteins as biomarkers in brain and metabolic disorders.

Diseases		EVs Derivation	Isolation Method	Protein Biomarker and Alteration in Disease	Detection Method	Study
**Neurodegenerative Disorders**	AD	Plasma	Polymer-Based Precipitation	Synaptophysin↓ Neurogranin↓ GAP43↓ Synapsin 1↓	ELISA	[222]
		Serum	Polymer-Based Precipitation	AACT↑ APOH↓	LC-MS	[223]
		Plasma	Ultracentrifugation	A0A0G2JRQ6↑ C1QC↑ CO9↑ GP1BB↑ RSU1↑ ADA10↑	LC-MS/MS	[224]
		Serum/Plasma	Polymer-Based Precipitation	P-T181-tau↑ P-S396-tau↑ Aβ1-42↑	ELISA	[225]
	PD	CSF	Ultracentrifugation	α-synuclein ↓	WB	[226]
		Plasma	Immunocapture/Ultracentrifugation	Tau↑	Simoa Assay	[227]
		Plasma	Polymer-Based Precipitation	DJ-1↑	ELISA	[228]
		Plasma	Ultracentrifugation	AChE↓	ELISA, WB	[229]
	ALS	Plasma	Ultracentrifugation	SOD1↑ TDP-43↑ FUS↑	WB	[218]
		CSF	Size Exclusion Chromatography	INHAT repressor↑	LC-MS/MS	[230]
		CSF	Ultrafiltration Liquid Chromatography	BLMH↓	LC-MS/MS	[231]
		Plasma	Ultracentrifugation, Nickel-Based Isolation	HSP90↓	WB, LC-MS/MS, ELISA	[232]
		Plasma	Polymer-Based Precipitation	CORO1A↑ HNRNPD↑	WB, LC-MS/MS, ELISA	[233]
	Multiple Sclerosis	Serum/CSF	Polymer-Based Precipitation	MOG↑	ELISA/WB	[234]
		CSF	Ultracentrifugation	Fibronectin↑ GFAP↑	WB, LC-MS/MS	[235]
		Serum	Polymer-Based Precipitation	LMP1↑	FC, WB	[236]
**Neuropsychiatric Disorders**	Depression	Plasma	Polymer-Based Precipitation	Cyclophilin D↓ Mitofusin-2↓	ELISA	[237]
		Plasma	Polymer-Based Precipitation	IRS-1↑	ELISA	[238]
	SCZ	Plasma	Polymer-Based Precipitation	GFAP↑ α-II-Spectrin↓	WB	[216]
	ASD	Serum	Polymer-Based Precipitation	Total protein↑	WB	[217]
	TBI	Plasma	Polymer-Based Precipitation	Aβ42↑NRGN↓ postsynaptic protein↓	ELISA, Simoa Assay	[239]
		CSF	Ultracentrifugation	αII-spectrin↑ GFAP↑ UCH-L1↑	LC-MS/MS, WB	[240]
		Plasma	Polymer-Based Precipitation	tau↑ amyloid-beta 42↑ IL-10↑	Simoa Assay	[241]
	PTSD	Plasma	Polymer-Based Precipitation	1L10↑	Simoa Assay	[241]
		Plasma	Polymer-Based Precipitation	neurofilament light chain↑	Simoa Assay, ELISA	[70]
**Neural** **Tumor**	GBM	CSF	Ultracentrifugation	TDP-43↑	WB	[242]
**Metabolic Disorders**	T2D	Urine	Differential Ultracentrifugation	PEPCK↑	ELISA, WB	[219]
	DN	Urine	Differential Ultracentrifugation	AMBP↑ MLL3↑ VDAC1↓	LC-MS/MS	[243]
	DR	Urine	Differential Ultracentrifugation	Junction plakoglobin↑	LC-MS/MS, WB	[244]
	T1D	Urine	Differential Ultracentrifugation	Wilm’s Tumor-1↑	WB	[220]
		Plasma	Polymer-Based Precipitation	RAB40A↑ SEMA6D↑ COL6A5↑	LC-MS/MS, WB	[245]
	NAFLD	Serum	Differential Ultracentrifugation	APOC1↑ APOC3↑ HP↑	LC-MS/MS	[246]
	HC	Plasma	Polymer-Based Precipitation	C1QB↑ C1QC↑C4BPA↑ C4BPB↑	LC-MS/MS, WB	[247]
		Serum	TiO2 Enrichment Technology	TGFB1↑ VWF↑ LGALS3BP↑ FGG↓ FGA↓ FGB↓	LC-MS/MS	[248]

Abbreviations: AD, Alzheimer’s Disease; PD, Parkinson’s Disease; ALS, Amyotrophic Lateral Sclerosis; SCZ, Schizophrenia; ASD, Autism Spectrum Disorder; TBI, Traumatic Brain Injury; PTSD, post-traumatic stress disorder; GBM, glioblastoma; T2D, type 2 diabetes; DN, diabetes nephropathy; DR, Diabetic Retinopathy; T1D, type 1 diabetes; NAFLD, non-alcoholic fatty liver disease; HC, hepatocellular cancer; CSF, cerebrospinal fluid; Simoa, single-molecule enzyme-linked immunoassays; WB, Western Blot; FC, flow cytometry; LC-MS, Liquid Chromatography–Mass Spectrometry; LC-MS/MS, Liquid Chromatography–Tandem Mass Spectrometry; ELISA, Enzyme-Linked Immunosorbent Assay; ↑, increased abundance; ↓, decreased abundance.

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
