# Peer review of "Approaches and Challenges in Characterizing the Molecular Content of Extracellular Vesicles for Biomarker Discovery"

_biomolecules, 2024, doi:10.3390/biom14121599_

Round 1

Reviewer 1 Report

Comments and Suggestions for Authors

In this review, Lee et al. focus on the methods for analyzing RNA and protein molecules carried by EVs. They summarize the existing methods and their respective advantages and disadvantages (Tables 1 & 3). Furthermore, they compile papers on RNA molecules and proteins proposed as diagnostic markers in EVs, indicating the analytical methods used and the associated diseases (Tables 2 & 4). These four tables are exceptionally well-organized, making this review a very useful guide for researchers new to the EV diagnostics field. The authors' overview of the RNA species that are targets for analysis in EVs, preceding the introduction of RNA analysis methods, is also commendable.

The only concern is in the introduction, where the authors state, "EVs can be categorized into three major types: exosomes, microvesicles, and apoptotic bodies." This classification is an outdated concept used over a decade ago. It is now widely recognized that the classification of EVs is not so straightforward. Referencing more current sources, such as the "Minimal information for studies of extracellular vesicles (MISEV2023)" in J Extracell Vesicles 13, e12404 (2024), and presenting an updated overall picture of EVs at the beginning would make this review more valuable for educating newcomers to the field.

Overall, this review provides a comprehensive and well-structured overview of EV analysis methods and their applications in diagnostics, making it a valuable resource for researchers in the field.

Author Response

Dear Reviewer 1,

We sincerely appreciate your comments.

Please see the attachment for our responses to your comments.  We also provide our responses here as well.

  1. Summary

Thank you very much for taking the time to review this manuscript. Please find our detailed, point-by-point responses below, along with the corresponding revisions/corrections highlighted in red in the re-submitted file.

  1. Point-by-point response to Comments and Suggestions for Authors

Comments 1: The only concern is in the introduction, where the authors state, "EVs can be categorized into three major types: exosomes, microvesicles, and apoptotic bodies." This classification is an outdated concept used over a decade ago. It is now widely recognized that the classification of EVs is not so straightforward. Referencing more current sources, such as the "Minimal information for studies of extracellular vesicles (MISEV2023)" in J Extracell Vesicles 13, e12404 (2024).

 Response 1: Thank you for your valuable feedback on EV classification. We agree that the classification of EVs is not straightforward. Therefore, we have deleted the previous outdated concept of the classification of EVs.  We have then updated the text and highlighted revisions in red in the introduction, reflecting the latest understanding of EV heterogeneity per MISEV2023 guidelines. Our update covers a comprehensive overview of EVs that includes various EV subtypes: exosomes, small ectosomes, microvesicles, and apoptotic bodies, along with their sizes and biogenesis pathways, and incorporated MISEV2023's categorization of large EVs (>200 nm) and small EVs (<200 nm).

Comments 2: Presenting an updated overall picture of EVs at the beginning would make this review more valuable for educating newcomers to the field.

Response 2: Thank you for pointing this out. We appreciate this suggestion. Accordingly, we have created a new diagram, Figure 1, that reflects the latest understanding of EV classification. This figure includes an updated overall picture of EVs that contains both small EVs (exosomes, small ectosomes) and large EVs (microvesicles and apoptotic bodies), with distinctions based on biogenesis, size, and molecular composition.

Reviewer 2 Report

Comments and Suggestions for Authors

This review aims to provide an overview of the two major biochemical cargos, RNA and protein, and their emerging significance in biomarker development, as well as the technologies employed to characterize them. The authors highlight and discuss the benefits and drawbacks of various approaches, challenges in the field, and potential directions for future developments. Here are the main comments.

1.       While general methods to characterize RNA and proteins are described, it is not clear how each of these methods provide advantages and disadvantages when specifically characterizing EVs?

2.       The authors need to elaborate on the fact that significant advances have been made over the last 5 years in various EV isolation methods from human biofluids and each of these methods could provide different types of challenges and advantages to each RNA and protein characterization method.

3.       In Table 3, the authors are showing different advantages and disadvantages of different protein characterization methods. Was a same preparation of EV compared on all these different methods? It appears that these advantages and disadvantages are general for the different methods and do not reflect direct comparison on specific isolation of EV. In addition, many of the characteristics are not comprehensive. For example, a major disadvantage of WB is that it only allows examination of 1-2 proteins at a time. For MS methods, several inaccurate or outdated statements are presented. For example, new advances in MS instruments and MS methods have been made over the last 5 years that have made MS methods much more sensitive, with high multiplexing capability, requiring a lot lower amounts of biofluids, than other protein characterization methods, so statements such as “Lower sensitivity compared to WB”, “Usually require large amount of sample”, “Favorable to abundant proteins” are likely outdated and no longer accurate. The authors need to review the recent literature and advances in all the protein methods (even WB) and more comprehensively compare them. Presumably, the same could be said for the different RNA characterization techniques mentioned in Table 1.

Author Response

Dear Reviewer 2,

We sincerely appreciate your insightful comments.

Please see the attachment for our responses to your comments.  We also provide our responses here as well.

  1. Summary

Thank you very much for taking the time to review this manuscript and for providing for your valuable feedback and comments. Please find our detailed point-by-point responses below, along with the corresponding revisions/corrections highlighted in red in the re-submitted file.

  1. Point-by-point response to Comments and Suggestions for Authors

Comments 1: While general methods to characterize RNA and proteins are described, it is not clear how each of these methods provide advantages and disadvantages when specifically characterizing EVs?

Response 1: Thank you for pointing these out. Therefore, we have made revisions to clarify that these methods are not only generalized but also applicable to the characterization of EVs. Additionally, we have updated Table 2 to include additional advantages and disadvantages specifically related to EV analysis, with the latest references highlighted in red. We believe that this enhancement will enables a better understanding of EV cargo characterization

Comments 2The authors need to elaborate on the fact that significant advances have been made over the last 5 years in various EV isolation methods from human biofluids and each of these methods could provide different types of challenges and advantages to each RNA and protein characterization method.

Response 2: We appreciate your insightful comments. In response, we have added a new section (2. Key Methods for EV isolation), which briefly reviews EV isolation methods including advances made over the past five years. Furthermore, we have created a new table, Table1, summarizing the advantages and disadvantages of these various isolation techniques.

Comments 3:

Comment 3.1: In Table 3, the authors are showing different advantages and disadvantages of different protein characterization methods. Was a same preparation of EV compared on all these different methods? It appears that these advantages and disadvantages are general for the different methods and do not reflect direct comparison on specific isolation of EV.

Response 3.1: We appreciate your insightful comments. We have, accordingly, added changes to clarify that these methods are broadly employed and also applicable to the characterization of EV proteins. In Table 4, we have included additional advantages and disadvantages specifically related to EV analysis, with the latest references highlighted in red. Given the complexity and variability in EV isolation and preparation, as well as the inherent differences in each characterization method, a comprehensive experimental comparison seems beyond the scope of this manuscript. However, to address this point, we have added different EV isolation methods in Table 1 and the new section (2. Key Methods for EV isolation).

Comment 3.2: In addition, many of the characteristics are not comprehensive. For example, a major disadvantage of WB is that it only allows examination of 1-2 proteins at a time. For MS methods, several inaccurate or outdated statements are presented. For example, new advances in MS instruments and MS methods have been made over the last 5 years that have made MS methods much more sensitive, with high multiplexing capability, requiring a lot lower amounts of biofluids, than other protein characterization methods, so statements such as “Lower sensitivity compared to WB”, “Usually require large amount of sample”, “Favorable to abundant proteins” are likely outdated and no longer accurate. The authors need to review the recent literature and advances in all the protein methods (even WB) and more comprehensively compare them.

Response 3.2: We have updated the protein characterization section (4.1 Methods to characterize EV-associated proteins) and Table 4 to reflect new advances made in MS field, Western blotting, and other protein characterization methods. For example, The WB section has been thoroughly revised to provide up-to-date information on current applications and its limitations, especially in the context of EV analysis. We have addressed the previously underrepresented aspects of its sensitivity, throughput, and capacity to analyze only a few proteins at a time. These updates are reflected in the text between lines 426-433 and 438-445. 

We have also revised the section on MS to reflect recent advances in sensitivity, multiplexing capabilities, and the ability to analyze smaller amounts of biofluid, which have made MS methods more efficient and applicable to a wider range of protein characterization needs. These changes are supported by the most recent studies, reflecting improvements that have significantly advanced MS technology over the last five years. These updates can be found in lines 522-532.
Recent references have been incorporated to support these updates, and all newly added references are highlighted in red for easy identification. This ensures that Table 4 and the surrounding sections now reflect the most current research and technological progress in both MS and WB methods.

Comment 3.3: Presumably, the same could be said for the different RNA characterization techniques mentioned in Table 1.

Response 3.3: We modified the RNA characterization techniques in Table 2 to ensure methods are current and accurately represent their advantages and challenges, with updated references highlighted in red. Additionally, we have added some other relevant references throughout the text of the review, all of which are also highlighted in red.

We believe these revisions address your concerns and provide a more accurate and comprehensive comparison of both protein and RNA characterization techniques of EV cargos. Thank you again for highlighting these important points, which have helped improve the manuscript.

Round 2

Reviewer 2 Report

Comments and Suggestions for Authors

The authors have addressed the main comments that were raised.

Author Response

Comments and Suggestions for Authors: The authors have addressed the main comments that were raised.

Response: We sincerely appreciate you for taking time to review and make the insightful comments.

We believe we have adequately addressed the main comments and look forward to receiving a positive outcome soon.

Best wishes,

Kai and Inyoul

-------------------------------------

Institute for Systems Biology

401 Terry Ave. N. 

Seattle, WA  98109

--------------------------------------